# OpenReview forum: "Pruning for Robust Concept Erasing in Diffusion Models"
_NeurIPS.cc/2024/Workshop/SafeGenAi — SafeGenAi Poster_

### Official Review · Reviewer_ibjq · 2024-10-08
**The paper identifies why finetuning approaches regarding concept erasure are prone to adversarial attacks and resolve this using a mixed pruning-erasure based objective. Very well funded, novel arguments that are logically treated and resolved.**

**Rating:** 8
**Confidence:** 4

**Review:**

Summary:
The authors notice that finetuning based concept erasure methods are flawed, as they only deactivate certain diffusion paths without entirely removing them. To resolve this, they propose a pruning method to completely remove these problematic paths. The pruning strategy only requires masking very few of the weights (around 0.001%). The masks are trained simultaneously with a free to choose widespread concept erasure method.

Strengths:
 - Clear explanation of why finetuning methods fail when adversially attacked (highly relevant)
 - Flexible training objective that can be used with a free-to-choose concept erasure objective
 - Strong results with very few weights masked, therefore maintaining most of the model capacities unharmed
 - Very self-critical paper, clearly stating and sicussing the possible limitations of the approach, much appreciated

Weaknesses:
 - No clear objections

General comment:
I am surprised by how few of the weights need to be altered. I would be afraid that other adversarial paths undetected by current attack methods still exist, and that also removing these would require changing significantly more weights. I would have thought that there were "more paths to Rome", an analysis on that regard (even beyond the safety perspective) would be very valuable.

Review summary:
The paper identifies why finetuning approaches regarding concept erasure are prone to adversarial attacks and resolve this using a mixed pruning-erasure based objective. Very well funded, novel arguments that are logically treated and resolved.

---

### Official Review · Reviewer_9RKX · 2024-10-09
**The paper presents an effective pruning-based approach to enhance the robustness of concept erasure in diffusion models, showing significant improvements over existing methods but lacking deeper theoretical analysis and broader generalization.**

**Rating:** 5
**Confidence:** 1

**Review:**

This paper proposes a pruning-based framework to improve the robustness of concept erasure in text-to-image diffusion models. The approach addresses the limitations of current fine-tuning-based methods by pruning key parameters in the neural network responsible for generating undesirable content, such as NSFW images or copyrighted artwork. The framework integrates both concept erasing and pruning into a unified objective, effectively preventing the reactivation of erased concepts under adversarial prompts. The experiments show significant improvements over state-of-the-art concept erasing techniques, particularly in resisting adversarial attacks.

### Pros

- **Identifies and Addresses Key Gap**: The paper successfully pinpoints a critical flaw in current concept erasing methods—their vulnerability to adversarial prompts. By introducing pruning to concept erasure, the authors provide a clear solution that tackles this problem head-on. It’s commendable that the authors took a widely known issue and designed a method to mitigate it specifically in the context of diffusion models.
- **Novel Use of Pruning in Concept Erasure**: Applying pruning techniques to concept erasure in generative models is innovative and demonstrates a novel application of an established method, commonly used in classification tasks, to generative models. This novel angle makes the approach stand out from other pruning applications.
- **Practical and Efficient Method**: The pruning-based approach manages to remove concept-related pathways while maintaining the model’s performance. By pruning less than 0.01% of parameters, the method shows both efficiency and effectiveness, a strong point for real-world deployment where models must remain robust while minimizing computational overhead.

### Cons

- **Limited Theoretical Depth**: While the method is effective, the theoretical reasoning behind why specific pruned parameters lead to robust concept erasure is not fully explored. The lack of detailed theoretical backing weakens the paper’s contribution, as readers are left to speculate about the underlying mechanisms of robustness.

---

### Official Review · Reviewer_gM9B · 2024-10-12
**This work addresses a critical challenge in text-to-image diffusion models: generating inappropriate or copyrighted content, especially when faced with adversarial prompts. The authors propose a pruning-based, variant method of ESD, which is simple and making sense. Comprehensive experiments were conducted to demonstrates this strategy's effectiveness.**

**Rating:** 8
**Confidence:** 4

**Review:**

### Strengths:
1. **Integration of Pruning and Erasing at Once**: The proposed strategy is an innovative approach ensuring concept erasing (especially for adversarial cases) by severing hidden state pathways, instead of directly finetuning on parameters. They also show that either pruning before erasing, or pruning after erasing works generally worse than integrating them into one single stage.

2. **Comprehensive Experiments**: The authors compare their method against multiple baselines, demonstrating its effectiveness.
---
### Suggestions:
1. A bit more Illustrations need to be added in the title of Figure 3, explaining what this figure shows, and I think this figure should be put on the top of page 5 or it might confuse readers. (before "Fig. 3 reveals the following mechanism: concept-erasing methods..." (line 151-158))
2. In table 3, "UnlearnDiff+Garbage Truck", there is a lack of explanation about why P-ESD works worse than ESD
3. It would be better to include how different levels of pruning affect robustness or generation quality in appendix.
---

### Overall:
I like this idea which is simple and working nicely. The experiments substantiate the effectiveness of your approach.